# 2,5-Dimethylfuran Production by Catalytic Hydrogenation of 5-Hydroxymethylfurfural Using Ni Supported on Al_2_O_3_-TiO_2_-ZrO_2_ Prepared by Sol-Gel Method: The Effect of Hydrogen Donors

**DOI:** 10.3390/molecules27134187

**Published:** 2022-06-29

**Authors:** Jorge Cortez-Elizalde, Gerardo E. Córdova-Pérez, Adib Abiu Silahua-Pavón, Hermicenda Pérez-Vidal, Adrián Cervantes-Uribe, Adrián Cordero-García, Juan Carlos Arévalo-Pérez, Norma Leticia Becerril-Altamirano, Nayi Cristel Castillo-Gallegos, María Antonia Lunagómez-Rocha, Jorge Noe Díaz de León, Zenaida Guerra-Que, Alejandra E. Espinosa de los Monteros, José Gilberto Torres-Torres

**Affiliations:** 1Centro de Investigación de Ciencia y Tecnología Aplicada de Tabasco (CICTAT), Laboratorio de Nanomateriales Catalíticos Aplicados al Desarrollo de Fuentes de Energía y Remediación Ambiental, División Académica de Ciencias Básicas (DACB), Universidad Juárez Autónoma de Tabasco, Km. 1 Carretera Cunduacán-Jalpa de Méndez, Cunduacán 86690, Mexico; link-190@hotmail.com (J.C.-E.); enrique_cordova90@hotmail.com (G.E.C.-P.); adibab45@gmail.com (A.A.S.-P.); hermicenda.perez@ujat.mx (H.P.-V.); adrian.cervantes@ujat.mx (A.C.-U.); adrian.cordero@ujat.mx (A.C.-G.); carlos.arevalo@ujat.mx (J.C.A.-P.); norma.becerril00@gmail.com (N.L.B.-A.); nayicastillogallegos15@gmail.com (N.C.C.-G.); alejandra.espinosa@ujat.mx (A.E.E.d.l.M.); 2Centro de Nanociencias y Nanotecnología, Universidad Nacional Autónoma de México, Km. 107 Carretera Tijuana-Ensenada, Apdo. Postal 14, C.P., Ensenada 22800, Mexico; noej@ens.cnyn.unam.mx; 3Laboratorio de Investigación 1 Área de Nanotecnología, Tecnológico Nacional de México, Km. 3.5 Carretera Villahermosa–Frontera, Cd. Industrial, Villahermosa 86010, Mexico; zenaida.gq@villahermosa.tecnm.mx

**Keywords:** 5-hydroxymethylfurfural, 2,5-dimethylfuran, hydrogen donor, Ni^0^NiO/ATZ3_WI_ catalyst, sol-gel method

## Abstract

5-Hydroxymethylfurfural (5-HMF) has been described as one of the 12 key platform molecules derived from biomass by the US Department of Energy, and its hydrogenation reaction produces versatile liquid biofuels such as 2,5-dimethylfuran (2,5-DMF). Catalytic hydrogenation from 5-HMF to 2,5-DMF was thoroughly studied on the metal nickel catalysts supported on Al_2_O_3_-TiO_2_-ZrO_2_ (Ni/ATZ) mixed oxides using isopropanol and formic acid (FA) as hydrogen donors to find the best conditions of the reaction and hydrogen donor. The influence of metal content (wt%), Ni particle size (nm), Nickel Ni^0^, Ni^0^/NiO and NiO species, metal active sites and acid-based sites on the catalyst surface, and the effect of the hydrogen donor (isopropanol and formic acid) were systematically studied. The structural characteristics of the materials were studied using different physicochemical methods, including N_2_ physisorption, XRD, Raman, DRS UV-Vis, FT-IR, SEM, FT-IR Py_ad_, H_2_-TPD, CO_2_-TPD, H_2_-TPR, TEM and XPS. Second-generation 2,5-DMF biofuel and 5-HMF conversion by-products were analyzed and elucidated using ^1^H NMR. It was found that the Ni^0^NiO/ATZ3_WI_ catalyst synthesized by the impregnation method (WI) generated a good synergistic effect between the species, showing the best catalytic hydrogenation of 5-HMF to 2,5-DMF using formic acid as a hydrogen donor for 24 h of reaction and temperature of 210 °C with 20 bar pressure of Argon (Ar).

## 1. Introduction

The great environmental impact caused by the enormous consumption of resources derived from oil has attracted much attention, especially the problem of greenhouse gases generated by fossil fuels. This enormous environmental problem can be solved by taking advantage of renewable resources such as versatile liquid biofuels produced through lignocellulosic biomass. Lignocellulosic biomass has stood out because it is the most abundant and economical renewable raw material. This residual biomass is mainly composed of a high percentage of cellulose. The acid hydrolysis of cellulose leads to the extraction and conversion of glucose (C_6_H_12_O_6_) into platform molecules such as 5-Hydroxymethylfurfural (5-HMF); this 5-HMF is the result of the dehydration of aldohexose [1].

5-HMF is described as one of the 12 key platform molecules derived from biomass by the US Department of Energy [2]; 5-HMF is recognized as a highly versatile intermediate that can generate liquid biofuels from second-generation molecules such as 2,5-dimethylfuran (2,5-DMF) and sustainable chemical products [3]. There are a wide range of high-value derivatives of this platform molecule, particularly 5-ethoxymethylfurfural (EMF), 2,5-dihydroxymethylfuran (DHMF), 1,6-hexanediol (HDO), 2,5-diformifuran (DFF) and acid 2,5-furandicarboxylic (FDCA). All these derivatives are created by various chemical reactions such as hydrogenation, oxidation, decarbonylation, etherification, etc., producing various chemicals that serve as basic components, including pharmaceuticals, polymers, resins, solvents and fuel additives for which they have received great interest [4,5] (Figure 1).

Research has shown that 2,5-dimethylfuran (2,5-DMF) can replace the use of fossil fuels in the coming years. The properties of this compound are of great value because it shows energy efficiency and low ecological impact supplying the carbon cycle from biomass. This second-generation biofuel would be more optimal than ethanol (EtOH) and as efficient as gasoline; 2,5-DMF (mp = 92–94 °C) has a higher boiling temperature than ethanol (mp = 78.37 °C), which can reduce fuel loss through evaporation and the risk of explosions from volatile gases [6]. Experts describe that this biofuel, which has an energy efficiency of 31.5 MJ/L, is 40% higher than ethanol (21.3 MJ/L). An additional advantage of 2,5-DMF is that it is insoluble in water; this property would prevent contamination due to the absorption of humidity from the environment since this absorption of humidity would generate a decrease in engine performance. Another important characteristic is its high antiknock power, greater than that of ethanol (RON = 119 > 110) [7].

2,5-DMF can also be used as an additive to improve the quality and efficiency of gasoline; therefore, a small ratio between the mixture of the additive compared to ethanol could be used to optimize the performance of the gasoline engine and release fewer pollutants. To further investigate the combustion and emission characteristics of 2,5-DMF in internal combustion engines, experiments of pure 2,5-DMF, 2,5-DMF/gasoline blends and 2,5-DMF/diesel blends have been carried out, respectively. All results indicate that this 2,5-DMF biofuel has the potential to become an effective alternative to gasoline [8]. The hydrogenolysis of 5-HMF to 2,5-DMF requires a bifunctional catalyst using molecular H_2_ with hydrogenation and deoxygenation activity. Noble metals have been used to conduct these necessary reactions, for instance, Pd, Pt, Ru and Au, which, at lower loadings or impregnation percentages, showed very high catalytic hydrogenation activity; however, these catalysts suffer from leaching due to the weak interaction between the support and the metallic nanoparticles. Catalysts based on transition metals (Ni, Co, Fe, Cu, etc.) have a wide range of catalytic applications that involve reactions such as hydrogenation, hydrogenolysis, hydrodeoxygenation, etc., [9]. Non-noble metals are easily available; as a result of their low cost and high load, they also possess suitable properties to produce 2,5-DMF [10], which favors their use with respect to noble metals (Pt, Ru, Pd, Au).

Nickel (Ni)-based catalysts on acidic supports have great prospects for the selective hydrogenolysis of 5-HMF to 2,5-DMF. Lima et al. reported that the synthesis of 2,5-DMF was catalyzed by Ni Raney catalyst; however, reaction by-products such as 2,5-dimethyltetrahydrofuran (2,5-DMTHF) and 2,5-dihydroxymethylfuran (2,5-DHMF) generated by the selective hydrogenation of 5-HMF were detected, obtaining low yields and selectivity of 2,5-DMF. This is due to the strong hydrogenation capacity of nickel [11]. Nevertheless, Kong et al. achieved the switchable synthesis of 2,5-DMF with a 96% yield over Ni Raney; the excellent yields can be explained by the fact that when modifying a reaction parameter such as the temperature increase (120 to 220 °C), deoxygenation of the hydroxymethyl group is promoted, producing high yields of this biofuel since nickel facilitates the hydrogenation reaction but has a limited deoxygenation capacity at low temperature [12]. Siddiqui et al. prepared nickel oxide (NiO) nanoparticles at different percentages (1–7% Ni in Ni/WO_3_) supported by nanostructured tungsten oxide, demonstrating that the activity of the catalyst as well as the high yield and selectivity of 2,5-DMF depend on the increase of nickel charge, producing selective hydrogenation of the platform molecule derived from biomass 5-HMF to 2,5-DMF and obtaining a 99% conversion of 5-HMF with 95% selectivity of 2,5-DMF in 6 h of reaction [10].

One of the very important surface properties of catalysts are the metallic and acid-basic active sites; these sites determine the selectivity path of the desired product, that is why bifunctional catalysts with these active sites are generally used for different reactions such as hydrogenation, catalytic cracking, simultaneous isomerization, etc. Zhu et al. evaluated a catalyst of Nickel impregnated in zirconium oxide modified with phosphate Ni/ZrP. This catalyst carried out hydrogenation in the presence of the metallic sites of nickel; likewise, it executed hydrogenolysis which took place on the Lewis acid sites of Ni/ZrP, caused by the zirconium vacancies of 5-hydroxymethylfurfural (5-HMF) to 2,5-dimethylfuran (2,5-DMF), achieving a yield of 68.1% of 2,5-DMF and a conversion of 100% of 5-HMF at 240 °C, 5 MPa of H_2_ and 20 h of reaction [13]. Kong et al. reported in one of their research works that the hydrogenolysis reaction at low temperature (particularly on non-noble metal catalysts) is a challenge. However, they synthesized nickel nanoparticles (NPs) at 36% of this metal with inlays of nickel phyllosilicate (NiSi-PS) and evaluated it at temperatures lower than 150 °C with H_2_ pressure generating 90.2% yield towards 2,5-DMF. The high yield was originated from a synergy between the highly dispersed nanoparticles and the active sites; despite that, the acidic sites generated by the nickel phyllosilicate structure of the NiSi-PS catalyst caused selective hydrogenolysis showing a conversion of 5-HMF of two times in hydrogenolysis and of three times in comparison to the impregnated Ni/SiO_2_, a previously investigated material [14]. Another Nickel catalyst was supported on Al_2_O_3_ derived from hydrotalcite NiAl-CT at different calcination temperatures (300 to 850 °C). These catalysts can efficiently and selectively convert 5-HMF into 2,5-DMF; this is due to the strong interaction of NiAl_2_O_3_ species that created metal and bifunctional acid–base sites on the surface, achieving simultaneous reactions of hydrogenation and hydrogenolysis but with selective hydrogenation to 2,5-DMF obtaining 91% of the yield of this biofuel [15]. Guo et al. described a monometallic catalyst of 10 to 50% *w*/*w* based on the Nickel of Ni/ZSM-5, where mixtures of Nickel phases (Ni^0^ and NiO) were found. This mixture of phases generated metallic sites (Ni^0^) and Lewis acid sites Ni^2+^ (NiO) on the catalyst surface, with a balanced synergy between these sites available for selective hydrogenolysis to the production of 2,5-DMF (96.2%) with H_2_ pressure after 7 h of reaction [1].

It is reported that, in the reactions of 5-HMF with H_2_ pressure, a lower yield and selectivity for 2,5-DMF is obtained; this is because it causes the additional reduction of the furan ring producing dimethyltetrahydrofuran (DMTHF), these are the main problems in the selective hydrogenation of 5-HMF to 2,5-DMF. However, a solution to this is the transfer of hydrogen from hydrogen donor molecules such as acids and alcohols. Typical hydrogen donors include alcohols, such as ethanol, 2-propanol and formic acid (FA) [16]. Yang et al. investigated the influence of formic acid as a hydrogen donor on a bimetallic catalyst Ni 2% *w/w* Co 20% *w/w* supported on C (NiCo/C), generating the catalytic hydrogenation/hydrogenolysis of 5-HMF to 2,5-DMF with higher selectivity and yield, obtaining 90% yield, at a temperature of 210 °C, of this biofuel; it was also concluded that the hydrogen donor does not generate the spontaneous hydrogenation of the furan ring [8]. Han et al. prepared and evaluated nickel-molybdenum sulfide catalysts with mesoporous alumina support Ni-MoS_2_/mAl_2_O_3_, and these catalysts were used for the hydrogenolysis of 5-hydroxymethylfurfural towards 2,5-DMF under mild conditions, 130 °C and 1 MPa H_2_; the catalyst 5Ni-7MoS_2_/mAl_2_O_3_ achieved a 95% yield of 2,5-DMF, and isopropanol can cooperate with molecular hydrogen to improve the yield of 2,5-DMF forming ethers that were easier to remove [17]. Jae et al. investigated catalytic transfer hydrogenation (CTH) using secondary alcohols as hydrogen donors (isopropanol) on ruthenium catalysts supported by Carbon 5% Ru/C, resulting in the selective conversion of 5-HMF to 2,5-DMF, obtaining a yield above 80% at 190 °C, under the pressure of N_2_ [18].

In this work, Ni/ATZ catalysts were prepared by Wet Impregnation (WI) and Suspension Method (SM) and characterized using several characterization techniques. The catalytic hydrogenation reaction of 5-hydroxymethylfurfural (5-HMF) from lignocellulosic biomass was investigated to obtain versatile liquid biofuels such as 2,5-DMF. This reaction was carefully studied using 2-propanol and formic acid as hydrogen donors.

## 2. Results and Discussion

### 2.1. Materials Characterization

The textural properties of the supports and Ni/ATZ monometallic catalysts were determined by N_2_ physisorption and were evaluated to measure B.E.T. areas, volume and pore diameter. The values obtained (S_BET_, P_D_ and V_BJH_) of the supports and monometallic catalysts synthesized by Wet Impregnation (WI) and Suspension Method (SM) are shown in Table 1, where it can be observed that the ATZ2 support with a surface area of 278.32 m^2^/g decreases the surface area value at 203.40 Ni^0^/ATZ2_SM_ m^2^/g when impregnating 15 wt% of Nickel. It was also reported that when impregnating nickel, the specific areas decrease due to the incorporation of Nickel species (Ni^0^, NiO) in the support [19].

Average pore diameters were calculated using the BJH method for mesoporous solids; the materials also suffered a decrease in P_D_. The ATZ2 support showed a P_D_ of 7.55 nm; when the nickel was deposited, the P_D_ values decreased to Ni^0^NiO/ATZ2_WI_ 6.76 nm, Ni^0^/ATZ2_SM_ 5.43 nm and NiO/ATZ3_WI_ 5.21 nm, this is due to the incorporation of the metal generating agglomeration of Ni species, blocking pore cavities.

Figure 1 show the isotherms of the supports (ATZ) and Ni/ATZ monometallic catalysts synthesized by WI and SM. All the materials showed type IV isotherms according to IUPAC [20], characteristics of a mesoporous material, meaning that the absorbates have pore sizes from 2 to 50 nm. Likewise, an increase in the amount adsorbed at intermediate relative pressures was demonstrated and occurred through a multilayer mechanism where pore filling was carried out. Given the phenomenon of capillary condensation called hysteresis, the catalysts showed two types of hysteresis loops of type H_2_, corresponding to uniform spheres and bottleneck shapes.

The XRD patterns of the ATZX supports and Ni/ATZX catalysts with 15 wt% nickel shown in Figure 2a display the XRD patterns of the supports synthesized by the sol-gel method and calcined at 550 °C. Characteristic signals of completely amorphous supports can be seen in the interval [2θ = 17.8°, 38.9°] with the interaction between TiO_2_-ZrO_2_ [21,22], this could be due to the ionic radii being very close which suggests that Zr^4+^ ions enter the TiO_2_ network, replacing Ti^4+^ [23]. Manríquez et al. reported that at percentages lower than 50% of TiO_2_-ZrO_2_, an amorphous behavior occurs because there is no crystallization at temperatures below 600 °C by the sol-gel method. This strong interaction between Ti-O-Zr is caused by the fact that Zr is entering the TiO_2_ network formed during the condensation of partially hydrolyzed alkoxide precursors [24,25]. In the ATZ1 and ATZ2 supports, peaks can be seen corresponding to the planes at 2θ = 40°, 46.13° (400) and 66.61° (440) (card JCPDS-ICDD No 75-0921) [26], which are characteristic of alumina gamma formation because the alumina content is higher in these supports. Figure 2b show the X-ray diffraction patterns of the Ni/Al_2_O_3_-TiO_2_-ZrO_2_ monometallic catalysts synthesized by the wet impregnation method, where mixtures of Ni^0^NiO phases are observed in all the catalysts due to the calcination treatment of 400 °C, exhibiting peaks at 44.39°, 51.71° and 76.27° corresponding to the planes (111), (200) and (220) of metallic Ni^0^ (JCPDS-ICDD No. 04-0850) and the peaks at 37.23° (111), 43.25° (200) and 62.93° (220) correspond to the NiO phase (JCPDS-ICDD No. 47-1049) with a face-centered cubic structure (FCC) for the nickel species of Ni^0^NiO/ATZX_WI_ catalysts. In the diffractograms of Figure 2c, the calcination effect in oxidative (air) and reducing (H_2_) atmospheres are compared to generate the Ni^0^ and NiO species. It is observed that mixtures of nickel phases are generated when subjecting the Ni/ATZX_WI_ catalyst to a reducing atmosphere (H_2_ flow), while in an oxidative atmosphere, only NiO species are generated, presenting crystalline planes in 2θ: 37.22 (111), 43.33 (200) 62.96 (220) and 75.50 (311), corresponding to NiO with a face-centered cubic structure (FCC) [27]. Similarly, corresponding peaks of gamma-alumina 2θ = 40°, 46.13° (400) and 66.61° (440) are detected. The diffractograms of the Ni/Al_2_O_3_-TiO_2_-ZrO_2_ monometallic catalysts synthesized by the suspension method are found in Figure 2d and only present metallic Ni^0^ phases showing the corresponding peaks and planes at 44.39° (111), 51.71° (200) and 76.27° (220) of metallic Ni^0^ with FCC structure, this is due to the complete reduction of Ni(NO_3_)_2_.6H_2_O on the catalyst surface, since it was maintained in the presence of Hydrogen from the pretreatment of the sample and in catalyst reduction. All monometallic nickel catalysts exhibit small signals of nickel interacting with alumina that correspond to nickel aluminates NiAl_2_O_4_ (JCPDS-ICDD No. 10-0339) with planes (311), (400) and (440) with spinel structure [28]. These signals are corroborated with the XPS spectra obtained for these catalysts.

The diffractograms, in addition to providing us with the crystalline phases of the monometallic nickel catalysts, also allow us to calculate the average size of the crystal using the Scherrer equation. Table 2 show the particle sizes obtained by intense peaks for Ni^0^ 44.52° (111) and NiO 43.25° (200) of the XRD diffractograms. Average crystal sizes range from 5 to 19 nm, depending on the synthesis method used for the Ni/ATZ catalysts.

The Raman spectra (100 to 1000 cm^−1^) that were observed in Figure 3a correspond to the ATZ mixed oxides synthesized by the sol-gel method and Ni/ATZX_WI_ monometallic catalysts synthesized by the wet impregnation method. The Raman spectra of the mixed oxides (ATZ) did not show crystallinity, and they were characteristic of amorphous materials, as mentioned in the XRD analyzes. These amorphous solids tend to show a wide bandwidth instead of narrow peaks. Titania showed bands of low activity in the 200–600 cm^−1^ intervals that belong to amorphous TiO_2_ [29]. Meanwhile, there were no zirconia phase signals detected in the XRDs. A broad band that begins at 700 cm^−1^ and ends at 1000 cm^−1^, characteristic of amorphous ZrO_2_ [30], was found with Raman spectroscopy. For alumina, there were no defined signals, nor for phases of titania (anatase, rutile or brookite). In XRD, alumina presented characteristic bands of γ-Al_2_O_3_; however, in Raman spectra, there was an absence of signals because amorphous alumina exhibited very weak bands due to the low polarization of the light atoms and the ionic character of the Al-O bonds [31].

Figure 3b corresponds to the Raman spectra of the monometallic Ni/ATZX_WI_ catalysts synthesized by the wet impregnation method with mixtures of Ni^0^/NiO phases show characteristic peaks of Ni-O vibration modes where the peaks below 600 cm^−1^ are due to first-order scattering, and those above 600 cm^−1^ are due to second-order scattering [32]. The four bands correspond to FCC of NiO with vibrational modes 1TO (at >370), 1LO (at >570 cm^−1^), 2TO modes (at >730 cm^−1^), TO + LO (at >906 cm^−1^) and 2LO modes (at >1090 cm^−1^), respectively [33]. The signal of the band located at 381 cm^−1^ corresponds to the modes of vibration that are attributed to a first-order transverse optical (1TO) and longitudinal optical (1LO) phonon (1P) and 560 cm^−1^, which are excitations of a single phonon that are more pronounced in nano-sized NiO powders [34]. Likewise, they present Ni^2+^/Ni^3+^ (Ni_1-x_O) mixed valences corresponding to the dark color of monometallic catalysts [35]. Weak bands are found at > 704 cm^−1^ of the modes of vibration and are attributed to two transverse optical phonons (2P) of the second-order 2TO [36]. No signals were observed from bands at 925 cm^−1^ (TO + LO), 1100 cm^−1^ (2LO) and magnon bands (2M) at >1400 cm ^−1^ for NiO. This is perhaps because there was no collective excitation of electron spins in nickel nanoparticles. In the Raman spectra, vibratory bands that indicate nanoparticles of metallic Ni^0^ were not observed, as mentioned in the XRD analyzes [37].

The UV-Vis DRS spectra of the ATZX supports are shown in Figure 4a. Al_2_O_3_ absorption bands were not observed because alumina normally has an absorption edge close to 179 nm and is outside the region limit of the UV-Vis-DRS equipment [38]; therefore, it is considered an insulating material. The absorption edge in these ternary materials was located at approximately 350 nm, observing high absorption bands from 218 to 290 nm of titanium cations Ti^4+^ [39,40] with strong interaction between TiO_2_-ZrO_2_. Considering that titanium is a semiconductor, greater electronic transitions are generated, producing a binary system TiO_2_-ZrO_2_. This binary system can result from its isovalent states, electronegativity, and ionic radii [41]. Resonance of nanoparticles (Nps) is observed by generating the surface plasmon using UV-Visible DRS spectroscopy in the spectrum range from 200 nm to 800 nm in the catalysts synthesized by the IH and MS methods to confirm the formation of nanoparticles (Nps) of metallic nickel on the surface of the monometallic catalyst. The UV-Vis DRS spectra of the Ni/ATZX_SM_ nickel monometallic catalysts are shown in Figure 4b–d. Nickel species have various signals ranging from 350 to 700 nm [42]. The signal at 500 nm corresponds to the absorption of the surface plasmon due to the effect of resonance, related to the formation of metallic nickel nanoparticles (Ni^0^) [43]. It is known that the wavelength of the surface plasmon resonance is sensitive to several parameters such as the size and shape of the particles. The resonance band widens with the increase in the particle size of metal [44], and these bandwidths are produced by the metallic loading of Nickel (15 wt%) and are related to the Ni^0^ particle sizes obtained by the Scherrer equation between 5 to 9.8 nm by XRD as described in Table 2. The UV-Vis DRS spectra of Figure 4c correspond to the oxidized monometallic NiO/ATZ1_WI_ catalysts. The coordination and oxidation state of the nickel oxide (NiO) species are observed starting with a strong absorption around band 241 nm assigned to the O^2−^→Ni^2+^ charge transfer transitions [45]. A weak absorption band is observed that goes from 400 to 500 nm, and it is associated with the d→d transitions of Ni^2+^ ions, suggesting the presence of Ni^2+^ ions in tetrahedral coordination [46]. Weak bands at approximately 600 to 700 nm are attributed to the octahedrally coordinated Ni^2+^ species in the NiO lattice [47]. Ni^2+^ nickel ions are in a coordinated form in tetrahedral and octahedral sites, commonly associated with the partial formation of the nickel aluminate phase with a spinel-like structure in the lattice, NiAl_2_O_4_ [48]. This is also seen in the XPS spectra.

Figure 5 show the bands generated by the different vibrations of bonds in the materials. As shown in the Figure 5a the FTIR spectra supports, the signals of the found bands are ν: 3461.14 cm^−1^ OH, δ: 1636.6 cm^−1^ H–O–H, ν: 2376.4 C–H or CO_2_, ν: >800 Μ−O−Μ. Likewise, Figure 5b shows the infrared FT-IR spectra of Ni^0^NiO/ATZ1_WI_ are displayed before calcination in the Nickel impregnation from the precursor salt Ni(NO_3_)_2_.6H_2_O in the Al_2_O_3_-TiO_2_-ZrO_2_ supported by the WI method. Defined bands can be observed at wavenumbers ν: 3461.14 cm^−1^ OH and δ: 1636.6 cm^−1^ of the HOH attributed to the intrinsic humidity of the salt by the six water molecules that nickel nitrate has in its structure and synthesis method. The vibration modes of 1636.6 and 1383.9 cm^−1^ were attributed to asymmetric and symmetric vibrations of nitrates [49]. These nitrate anions interact with the support. After calcination at 400 °C, the nitrates are thermally decomposed, the bands mentioned above practically decrease and others disappear completely. This is produced by the elimination of nitrates and intrinsic moisture released in the form of water vapor. Bands are observed in modes of vibrations of ν: 3461.14 cm^−1^ OH and δ: 1636.6 cm^−1^ HOH. The introduction of Ni metal decreased the intensity of the band and two shoulders located between 900 to 400 cm^−1^, assigned to the vibration modes M–O, M–O–M and O–M–O (M=Ni) [50] with structures corresponding to the Ni–O bond of nickel oxide [51] of octahedral groups NiO_6_ in the face-centered cubic structure (FCC) [52]. These signals were also found in XRD and UV-Vis DRS; no signals were found for metallic nickel.

The reducibility and interaction between the Nickel species are detailed in the thermogram. This analysis was carried out for fresh catalysts with the impregnation of 15 wt% of Ni. The reduction profiles in Figure 6 show three peaks.

The first very sharp peak on α at 300 °C corresponds to the generated NiO particles. Ni^2+^ is reduced in a single stage to metallic nickel according to the following equation: NiO + H_2_ → Ni + H_2_O [53]. This direct reduction of free NiO to metallic Ni species, regardless of higher temperature, implies the highly stabilized and dispersed NiO species [54] due to a weak interaction on the surface of the support.

The β peak that occurs at a temperature of 400 °C is attributed to the reduction of the bulk of NiO (NiO-free species) [55,56]. The γ peak at 500 °C is due to the interaction of NiO species with different interactions with the tetrahedral coordination sites or the octahedral coordination sites of the support with the support at elevated temperatures [57]. The metallic nickel described in the XRD, UV-Vis DRS characterizations has been described in detail; however, there are other compounds formed by the interaction of nickel due to the conduction of oxygen ions through the support layer to the Al_2_O_3_-TiO_2_-ZrO_2_ interface; therefore, more species are generated (Ni^2+^) as NiO rather than Ni metallic on the surface of the catalyst [58]. This number of species is described in the XPS.

The H_2_-TPD profiles of the Ni/ATZ monometallic catalysts synthesized by the Wet Impregnation Method are shown in Figure 7a. Two types of H_2_ desorption peaks can be distinguished in the catalysts. The low-temperature peaks (below 100 °C) are attributed to hydrogen weakly bound to nickel active sites [59,60]. The second peak emerging from 200 to 500 °C is attributed to desorbed hydrogen located on the surface layers of the support. In this peak, there is greater desorption of hydrogen strongly bound to the surface of the catalyst [61]. The total amount of desorbed hydrogen was calculated from the deconvolution of the peak areas of the H_2_-TPD profiles, assuming a H/Ni: 1 stoichiometry that a hydrogen atom is adsorbed on a single nickel atom. With respect to this, the metallic surface area and percentage of dispersion were calculated. Table 3 display the calculated results of the areas of the H_2_-TPD profiles. The Ni^0^NiO/ATZ3_WI_ catalyst has a higher amount of H_2_ desorption (μmol H_2_/g_cat_) and a higher percentage of metallic dispersion than the other Catalysts. This desorption and metallic dispersion are attributed to the Ni metallic species present on the catalyst surface; however, it can be interpreted that there is a small ratio of metallic sites available on the catalyst surface, corroborated by the percentage of Ni^0^ species found in XPS. These metallic sites are necessary for the hydrogenation reaction to obtain biofuel.

Figure 7b show the CO_2_-TPD desorption profiles of the Ni^0^NiO/ATZX_WI_ catalysts, while the basicity characteristics of the catalyst are summarized in Table 4.

The sites are classified according to their basic strength related to the temperature desorption peaks of CO_2_, as described in the table. Desorption peaks are found at low temperatures near 100 °C and are attributed to the desorption of CO_2_ adsorbed on nickel on the catalyst surface [61], which is related to very weak basic sites of the catalyst, as can be seen in Table 4. These weak sites are related to CO_2_ molecules that interact with hydroxyl groups on the surface. Low-temperature peaks (100–450 °C) are identified as α sites and are attributed to CO_2_ molecularly adsorbed from metallic nickel [62], and the peaks at 250 °C of CO_2_ desorption are attributed to weak and basic intermediate sites. This basic site ratio may become an important factor in the selectivity of 2,5-DMF.

The FT-IR Py_ad_ analysis was carried out, which determines the qualitative and quantitative acidity of the Lewis and/or Brønsted acid sites present on the catalyst surface through the adsorption of vibrational gaseous pyridine. It was measured in the infrared spectral region of 1700–1400 cm^−1^. Figure 8 reveal the vibratory stretch bands of the pyridine ring in the 1700–1400 cm^−1^ region [63]. In the FT-IR Py_ad_ spectra at different temperatures Figure 8a and at 50 °C in the ATZ3 catalyst, intense bands can be observed for Lewis-type acid sites. On the other hand, the characteristic bands of Brønsted-type acidity are very small; consequently, when increasing the temperature from 100 to 400 °C, the Brønsted basic sites disappear, concluding that they represent a force of very weak sites. This strength of acidity increases through temperatures, having sites with weak (25 to 50 °C), medium (100 °C), intermediate (200 °C), strong (300 °C) and very strong forces of acidity (400 °C). Examining the vibrations of the pyridine ring with the surface sites of the monometallic catalyst with Ni^0^ metal species and NiO oxidized phases of nickel in the FTIR spectra shown in Figure 8b, all the materials exhibit the pattern of the bands with the vibrational modes of stretching for pyridine adsorbed upon interaction with an acid surface site at ν8a = 1605, ν8b = 1575, ν19a = 1488 and ν19b = 1447 cm^−1^ [64], characteristic of Lewis acid sites [65]. The bands with stretch vibrations ν8a = 1605 and ν19b = 1447 cm^−1^ are related to the coordinated LPy adsorbed on Lewis acid sites with the metals and cations present [66]. The bands ν8b = 1575 and ν19a = 1488 cm^−1^ relate to the LPPy of the pyridine electron pair with the M–O species. The intensity of the ν19b = 1447 cm^−1^ band is related to the surface concentration of pyridine and strong Lewis sites, while the ν8b = 1575 cm^−1^ band is assigned to weak Lewis sites. Pyridine FT-IR spectra do not show characteristic signals, indicating the generation of acidic Brønsted sites for any catalyst above 100 °C. As mentioned above, at high temperatures, weak Brønsted sites disappear.

The number of acid sites characterized by FT-IR Py_ad_ was calculated from the areas under the curves and related to the pyridine concentration according to the procedure conducted by C.A. Emeis in 1993. The results are shown in Table 5. The total acidity present in the supports and monometallic catalysts for the ATZX supports indicates that the amount μmol_py_/g_cat_ decreases as the relationships in the Al_2_O_3_, TiO_2_ and ZrO_2_ vary. While the decrease of Lewis sites is appreciated in the impregnation of nickel, these Lewis sites are generated by the species in the monometallic catalysts NiO: 160 μmol_py_/g_cat_ > Ni^0^ NiO: 146 μmol_py_/g_cat_ > Ni^0^: 108 μmol_py_/g_cat_, obtaining that the NiO/ATZ3_WI_ catalyst contains more Lewis sites per Ni^2+^ of the oxidized species. In contrast, the other Ni^0^ and Ni^0^NiO species contain fewer Lewis sites, which may be due to the reduction of NiO species to metallic sites (Ni^0^), as observed in the H_2_-TPD and H_2_-TPR.

The qualitative chemical mapping by EDS in Figure 9 shows the dispersion of the elements present in the ATZ3 catalyst. Good dispersions are observed for elements such as titanium and zirconium, which are highly dispersed in the matrix of the mixed oxide synthesized by the sol-gel method; these distributions agree with those reported by the synthesis method [67]. Aluminum shows a good dispersion on the mixed oxide matrix but reveals some areas with agglomerations. This is due to small crystallization of the oxidized aluminum in the catalyst, which is correlated with the appearance of some small peaks for Al_2_O_3_ in the XRD diffractograms described above. Furthermore, Figure 9 present qualitative analysis by the EDS detector of the dispersion of 15 wt% Ni in the ATZ support. In catalysts with Ni^0^NiO/ATZ3 nickel phase mixtures, some agglomerations caused by the percentage of impregnated metal are observed [56]. These agglomerations are the result of large particles around 19 nm described in the XRD analysis. For the Ni^0^/ATZ3_SM_ catalyst, better dispersion of nickel particles can be observed. This is due to the suspension method used as a synthesis method that generates particles smaller than 10 nm.

Figure 10a show the TEM images of the ATZ3 materials at 550 °C. In the images of the ATZ samples, the amorphous matrix of the Al_2_O_3_–TiO_2_–ZrO_2_ supports can be observed at 50 nm. Electron diffraction did not show signs of crystallinity planes of any phase for the Al_2_O_3_–TiO_2_–ZrO_2_ system, which is why it is concluded that it is a completely amorphous material at 550 °C with strong interactions as could be observed with the XRD and Raman characterization techniques. Figure 10b exhibit the TEM image in bright field at 200 nm that corresponds to the monometallic catalyst synthesized by the suspension method Ni^0^/ATZ3_SM_. Semi-spherical Nickel nanoparticles can be observed on the external surface of the ATZX catalyst as well as the size of the particles, which varies between 8 and 9 nm, as shown in the histogram. It is important to emphasize that the sizes are similar to the results obtained by XRD, ranging from 5 to 9 nm. Despite the 15 wt% Nickel loads, a uniformity of these nanoparticles is noticed, which may be due to the suspension synthesis method that avoids particle agglomeration. Likewise, values of the interplanar distances 0.203 nm, 0.176 nm and 0.123 nm were obtained, coinciding with the planes (111), (200) and (220) of metallic Ni. These Ni^0^ species were also found in the X-ray diffractograms of Ni metallic with FCC structure [68]. Figure 10c display the monometallic catalyst with two nickel phases, metallic Ni^0^ and oxidized NiO. Nanoparticles with different hemispherical and hexagonal shapes with particle sizes that vary from 20 nm to 50 nm are recognized. This is due to the charge of nickel impregnated, which was 15 wt% Ni. The hexagonal-shaped crystal represents the NiO oxidation phase. The formation of hexagonal-shaped particles could be attributed to the hindered growth in the direction of some axes due to the adsorption of NO_3_^−^ resulting from the decomposition of the precursor Ni(NO_3_)_2_.6H_2_O during calcination at 400 °C on the faces (111) of NiO to form larger particles [69]. In addition, a space is observed between two adjacent planes with distances and their corresponding planes of Ni^0^ (111) 0.203 nm and (200) 0.176 nm, NiO (111) 0241 nm, (200) 0.208 nm and (220) 0.146 nm of the oxidized phase of Ni with face-centered cubic phase FCC [70]. Finally, a corresponding distance and plane of the NiAl_2_O_4_ (311) 0.243 nm species were found. These diffractions were also discovered in the XRD patterns for this catalyst. In Figure 10d, the monometallic catalyst with only the oxidized NiO phase is shown.

Semi-spherical NiO nanoparticles with an average size of 14 nm can be seen the in bright field micrograph. Although a hexagonal crystal of this oxidized nickel phase is also observed with a size between 18 to 20 nm, this crystal size is related to the particle size calculated by XRD, 19.6 nm, for this catalyst. The distances between two adjacent planes. (111) 0.241 nm, (200) 0.208 nm and 0.146 nm (220), corresponding to the oxidized phase of the NiO face-centered cubic phase (FCC), were located and calculated. In a similar way, 0.243 nm (311) and 0.201 nm (400) of the interaction of nickel with alumina formed NiAl_2_O_4_.

The XPS spectra at the Ni2p_3/2_ core level in Figure 11 correspond to the NiAZT3 monometallic catalysts prepared by WI and SM. All catalysts with different nickel phases (Ni^0^, Ni^0^NiO and NiO) exhibit a main peak at 851–858 eV and a satellite peak at a higher energy junction (BE) of 860–869 eV. These spectra indicated that Ni appeared in various oxidation states from the deconvolution of the main Ni 2p_3/2_ peak that generated three peaks, 852.4, 855.1 and 856.5 eV, for Ni^0^ and Ni^2+^, respectively, [71]. Figure 11a show the Ni^0^/ATZ3 catalyst. For nickel in the Ni^0^ metallic state, a small peak is seen at 852.4 eV [72]. For the Ni^2+^ oxidation state, signals were found with the compounds at 855.1 eV Ni(OH)_2_ with a satellite peak at 861.2 eV that can be attributed to the hydroxyl groups on the surface with nickel [73] because Ni is a highly active metal and has the ability to react directly to atmospheric oxygen to form nickel oxide or nickel hydroxide [74]. Signals corresponding to nickel aluminates NiAl_2_O_4_ at 856.5 eV with a satellite peak at 862 eV were found. This compound was formed because the Ni^2+^ species can occupy and coordinate in octahedral and tetrahedral sites with the alumina γ−Al_2_O_3_ representing spinel structures of NiAl_x_O_y_ and NiAl_2_O_4_ [26]. Figure 11b exhibit the Ni^0^NiO/AZT3_WI_ monometallic catalyst synthesized by the wet impregnation method, where, precisely, nickel species were found in their metallic (Ni^0^) and oxidized (NiO) forms. When Nickel is in the Ni^0^ metallic state, a small peak is also seen at 852.7 eV, and for the Ni^2+^ oxidation state, signals were found with the compounds at 855.2 eV of NiO with a satellite peak at 861.2 eV and nickel aluminates NiAl_2_O_4_ at 856.5 eV with a satellite peak at 862 eV. Figure 11c show the NiO/ATZ3_WI_ monometallic catalyst. The NiO species recorded a signal at 855.4 eV with a satellite peak at 861.2 eV. Likewise, nickel aluminate NiAl_2_O_4_ was observed, which exhibited a characteristic peak at 856.5 eV and a satellite peak at 861.4 eV. This indicates the presence of NiO and nickel aluminate on the surface with Ni^2+^ ions belonging to nickel aluminate [75]. This NiO species was detected in most characterization techniques and this catalyst only generated oxidized nickel species because, during the synthesis, it was in an oxidizing atmosphere at 400 °C. The results of XPS confirm these Ni^0^ and NiO phases, as mentioned in the other XRD, Uv/Vis and Raman characterization techniques.

Table 6 indicates the percentage composition of the nickel species (Ni^2+^ and Ni^0^). It can be seen that there is a small proportion of Ni^0^ metal species in the monometallic catalysts of 9.2 and 6.3%. This is due to the strong interaction of Ni^2+^ with the alumina, forming a nickel aluminate spinel (NiAl_2_O_4_) and limiting the formation of metallic nickel [76], as corroborated by the results of H_2_-TPD, where the metallic dispersion of 7% corresponds to the metallic sites found on the surface of the catalyst. However, the amount of Ni^2+^ species is found in a huge proportion with different nickel compounds (Ni(OH)_2_, NiO and NiAl_2_O_4_). This vast amount of Ni^2+^ species corresponds to the Lewis sites available, as mentioned in the FT-IR Py_ad_.

### 2.2. Catalytic Tests

#### 2.2.1. Catalytic Activity in CTH of 5-HMF to 2,5-DMF

Sacrificial agents such as formic acid and isopropanol were used to produce 2,5-DMF. In the same way, Dodecane (C_12_H_26_) was used as an internal standard with the function of having precise measurements and, in some cases, optimizing the reaction due to its inertness; the evaporation of 2,5-DMF was avoided by monitoring the mass balance during the reaction [77]. Figure 12a reveal the effect of using this internal standard. When the Ni^0^NiO/ATZ2_WI_ catalyst was tested without the internal standard, a 20% yield of 2,5-DMF was generated. This was caused by the loss of matter generated from critical reaction conditions such as temperature and pressure. The effect of adding Dodecane to the reaction can be appreciated since a 46% yield of 2,5-DMF was obtained. Yang et al. used a molar ratio (2:1 mmol) between the internal standard and 5-HMF, which optimized the thermodynamic equilibrium in the reaction to produce 2,5-DMF [8]. However, when this internal standard is doubled, the yield drops to 24%, inhibiting the reaction to produce 2,5-DMF. The addition of the internal standard helped us to perform the reactions with the sacrificial agents (formic acid and isopropanol) for the nickel monometallic catalysts.

Figure 12b displays the obtained yields of 2,5-DMF using formic acid as a hydrogen donor. Monometallic catalysts with Ni^0^NiO/ATZX_WI_ phase mixtures were evaluated. We can see that the Ni^0^NiO/ATZ1_WI_ catalyst obtained a yield of 24%, Ni^0^NiO/ATZ2_WI_ 42% and Ni^0^NiO/ATZ3_WI_ 46% of 2,5-DMF with a conversion of 99% of 5-HMF. These close yields of Ni^0^NiO/ATZ2_WI_ and Ni^0^NiO/ATZ3_WI_ catalysts can be explained by the limited amount of Ni^0^ metal sites on the catalyst surface. As mentioned in the results of H_2_-TPD, the close amounts of chemisorbed H_2_ of 190 and 194 μmol of H_2_ can be observed, as well as the near yields obtained in the reaction. For the Ni^0^NiO/ATZ3_WI_ catalyst, it was observed that it had 9.2% of metallic nickel species (Ni^0^). This limitation is provided by the interaction of nickel (Ni^2+^) with other compounds. These metallic sites available on the surface of these monometallic catalysts are very important for the hydrogenation of 5-HMF to 2,5-DMF.

##### Effects of Nickel Load and Particle Size

The effect of Ni loading on the catalytic performance was investigated for the Ni^0^NiO/ATZ3_WI_ monometallic catalyst in the conversion of 5-HMF to 2,5-DMF, and the results are shown in Figure 13.

When the metal loading is 10 wt%, the biofuel yield is relatively low, obtaining 21% of 2,5-DMF. This may be due to the fact that at lower loadings, the amount of metallic sites is less, which does not allow the hydrogenation reaction of 5-HMF. In many reports, loadings between 5 to 50% nickel impregnation are used. These charge increases generate more metallic sites and yields of this biofuel. A relevant particularity in the yield to produce 2,5-DMF is the sensitivity of the size of the nickel nanoparticles in the Ni^0^NiO/ATZ3_WI_ monometallic catalyst to produce 2,5-DMF. In the same Figure 13, the effect of particle size with respect to the Nickel charge can be seen, obtaining a 46% yield of 2,5-DMF with a particle size of 19.6 nm, while with the smaller particle size of 8 nm, only 21% yield is generated at 2,5-DMF and a conversion of 99% of 5-HMF. Chen et al. demonstrated this effect of Nickel impregnation load using percentages between 5 to 30 et%. These values were in the adequate percentage range to obtain high yields of 2,5-DMF; however, when increasing loads greater than 40 wt% nickel in the catalysts, the yields and the selectivity of 2,5-DMF decrease [78].

##### Effect of the Nickel Species of the Ni/ATZ3 Monometallic Catalyst

The Ni/ATZ3X monometallic catalysts with 15% Nickel synthesized by two methods (WI and SM) presented different Nickel species (Ni^0^, Ni^0^NiO and NiO) found with the characterization techniques (DRX, Raman, UV-Vis DRS and XPS). In Figure 14, the effect of these nickel species can be appreciated.

Analyzing the graph, the catalyst with the mixtures of Ni^0^NiO/ATZ3_WI_ species generated the highest yield (46%) of 2,5-DMF and a conversion of 99% of 5-HMF compared to the others that contain distinct species of nickel. The XPS results for this catalyst showed that it is the one that contains the most metallic species, 9.2% and 75% of NiO. Thus, helps to have a synergy between the acid and metallic sites present in the mixtures of Ni^0^NiO species that are of utmost importance in the esterification and hydrogenation reactions to obtain this biofuel since they demonstrate excellent yields and selectivity [79]. For catalysts with just the Ni^0^/ATZ3_SM_ phase, just a 30% yield of 2,5-DMF prevailed. The analyzed information from the XPS for this monometallic catalyst showed that the percentage of metallic sites was 6.3%, which limits the hydrogenation of 5-HMF. Likewise, other compounds with species (Ni^2+^) appeared that generated other esterification reactions of formic acid in 5-HMF, producing lower yields towards 2,5-DMF. The catalyst with the oxidized species NiO/ATZ3_WI_ generated a low 11% yield of 2,5-DMF and the interactions observed in the XPS (NiO and NiAl_2_O_4_) of these oxidized species showed that there is Ni^2+^ that represents available Lewis sites on the surface of the catalyst. In addition, by FT-IR Pyad, it was possible to quantify the amount of Lewis sites of 146 μmol_py_/g_cat_; therefore, only formats would be generated as the result of the esterification of HCOOH with 5-HMF. Conversely, there is published research that mentions that this oxidized nickel NiO phase does not generate any yields of 2,5-DMF since it only forms high yields of the compound 2,5-bis(hydroxymethyl)furan (BHMF) and 5-methyl furfuryl alcohol (MFA) [10]. This low yield of 2,5-DMF may be because some species are being reduced due to the spillover effect; however, it must be taken into account that the contribution of Ni^0^NiO species and metallic-acid sites are necessary to obtain high yields and selectivities of this second-generation biofuel.

##### Effect of the Acid-Basic Sites of the Ni/ATZ3 Monometallic Catalysts

The metallic, acidic and basic sites present on the surface of solid catalysts can be very selective in reactions to obtain biofuels such as 2,5-DMF. As mentioned, different species of Ni were generated (Ni^0^, Ni^0^, NiO and NiO). These species generate active sites that are very important to explain what happens with the chemical reaction. Acidic sites promote esterification reactions, and Ni^0^ acts as a metallic site to develop hydrogenation, while Brønsted sites donate protons to compounds forming carbocations which are promoters for the molecule’s absorption and can produce coke by deactivating the catalyst. Figure 15a shows the effect of the amount of Lewis acid sites present in the Ni^0^NiO/ATZ catalysts. These Lewis sites must be adequate and optimal to produce 2,5-DMF. The number of sites generated by the Ni^2+^ species was counted by FT-IR Py_ad_, quantifying them from the spectra, resulting in strong Lewis sites in the catalysts with values of Ni^0^NiO/ATZ1_WI_: 411 μmol_py_ < Ni^0^NiO/ATZ2 _WI_: 220 μmol_py_ < Ni^0^NiO/ATZ3 _WI_: 146 μmol_py_. The results of yields based on these Lewis sites showed that the Ni^0^NiO/ATZ3 _WI_ catalyst with the least amount of Lewis sites generated the highest yield of 46% of 2,5-DMF. This is because there is a balanced contribution of these acid sites and limited metallic, active sites, as mentioned above. However, for the Ni^0^NiO/ATZ1 _WI_ catalyst, the lowest yield was generated, 24% of 2,5-DMF. the excess of Lewis sites creates many formate compounds, as observed in Figure 2 as the result of etherification with the sacrifice agent; these by-products do not promote the formation of the biofuel 2,5-DMF.

Figure 15b compare the different nickel species that generate metallic and acid-base sites which are responsible for the production of 2,5-DMF. As stated before, the Ni^0^NiO/ATZ3 _WI_ monometallic catalyst has 9.2% nickel metal species, and the rest in percentage is due to interactions with the support producing different Ni^2+^ species; however, it has a moderate amount of Lewis sites Ni^0^NiO/ATZ3 _WI_: 146 μmol_py_/m^2^ existing equilibrium of metal sites and Lewis acid sites that can control the distribution of suitable products in the conversion of 5-HMF to 2,5-DMF. Sun et al. concluded that the species of nickel (Ni^0^, NiO) and Copper (Cu^0^) present in the Ni-Cu/SBA15 catalyst played a synergistic role in high yields of 2,5-DMF [80]. The catalyst synthesized by the suspension method Ni^0^/ATZ3_SM_ contains fewer metallic sites and a lower amount of Lewis sites compared to phase mixtures. Therefore, less hydrogenation of 5-HMF to 2,5-DMF is generated. Despite that, more formate by-product yields are created because of esterification. Finally, the NiO/ATZ3 _WI_ catalyst contains pure Lewis sites that only generate a minimal amount of 2,5-DMF. This huge amount of Lewis sites is suitable for the formation of hydrogen donor formates (formic acid) that react to 5-HMF, and they inhibit the formation of 2,5-DMF.

#### 2.2.2. Catalytic Transformation by Hydrogenation (CTH) of 5-HMF over Different Hydrogen Donors

The transfer of hydrogen from hydrogen donor molecules such as acids and alcohols, including ethanol, isopropanol and formic acid (FA), has been studied by different research groups [16]. Hydrogenation in the presence of alcohols as hydrogen donors using metal catalysts has been studied. Because alcohols provide important advantages as sources of hydrogen, they can also serve as reagents and solvents. Different alcohols have been used for the conversion of 5-HMF to 2,5-DMF. It is reported that using primary alcohols such as methanol as H_2_ donors generates low selectivity and yield of 2,5-DMF [81], while high yields of 2,5-DMF are obtained using secondary alcohols such as isopropanol (2-propanol) because secondary alcohols are excellent H_2_ donors [18,82]. In this work, different hydrogen donors, in particular, formic acid and isopropanol, were compared. The results with isopropanol as a hydrogen donor are shown in Figure 16, where the yields of 2,5-DMF from the conversion of 5-HMF are exhibited. For the Ni^0^NiO/ATZ3_WI_ catalyst, a low yield of 11% of 2,5-DMF and a conversion of 99% of 5-HMF were obtained.

In other works, isopropanol is used as a solvent and hydrogen donor. In this one, 0.86 mL was added in a similar relation to formic acid to compare its versatility. This low yield may be because isopropanol was used as a limiting reagent in the reaction medium, generating a limited concentration of H_2_ for hydrogenation in the presence of the catalyst since it is a secondary alcohol that decomposes as a ketone and releases H_2_ together with di-isopropyl ether and very low amounts of propane. Likewise, the temperature also influences the increase in the yields of 2,5-DMF. The BHMF at T > 180 °C becomes 2,5-DMF. The Ni^0^NiO/ATZ3_WI_ catalyst has the sites (metallic and acid-base) necessary for the reactions that occur in the 2,5-DMF reaction.

Monometallic catalysts with Ni^0^NiO/ATZ3_WI_ phase mixtures in the presence of formic acid as a hydrogen donor are suitable for this system in the production of 2,5-DMF (Figure 2). This hydrogen donor (formic acid) can be decomposed by dehydrogenation to form H_2_ and CO_2_ and dehydration to form H_2_O and CO [83]. It was reported by Yang et al. that the stoichiometry of formic acid is very important because 3 mmol of H_2_ is needed for the conversion of 1 mmol of 5-HMF to 2,5-DMF, which indicates that 3 mmol of formic acid is required (FA). Yields of 2,5-DMF were less than 10% using 3 mmol; therefore, the ratio was increased (1:10) regarding 5-HMF and FA, obtaining better results with 20 mmol, as the investigations reported in the production of this Biofuel [8]. There was no leaching of the nickel metallic phase in all the carried-out reactions due to the strong interaction of these metallic particles with the ATZX support.

The interaction of these metal and acid-base sites for the production of 2,5-DMF is explained in detail next: 2,5-DMF is produced from the hydrogenation, esterification and decarboxylation of 5-HMF, as indicated in Figure 2. The basic sites adsorb and activate the sacrificial agent formic acid (FA) on the catalyst. The surface Lewis acid sites have the function of adsorbing oxygenates such as 5-HMF, while the metallic sites adsorb and activate the hydrogens generated by FA, producing the catalytic hydrogenation of the C=O of the aldehyde, generating C–OH bonds to BHMF. Consequently, the contribution of the acid-base sites that absorb FA and 5-HMF generate formate ester reactions in the hydroxyl group OH including FHMF, FBHMF and FMFOL. These Lewis acid sites participate in the deoxygenation of formate esters FHMF to MFAL, where the hydroxyl group is reduced and interacts with adsorbed AF forming formate esters and resulting in the formation of FMFOL [84,85]. Lewis acid sites induce FMFOL deoxygenation to finally form the 2,5-DMF product [86]. However, the excess amount of hydrogen generated by the sacrificial agent can cause hydrogenation of the ring in 2,5-DMF and produce DMTF.

Figure 3 show the reaction scheme using isopropanol as a hydrogen donor. The reaction routes where it is observed that the metallic and acid-base surface sites of the catalyst cause the formation of 2,5-DMF. Jae et al. proposed a reaction scheme where isopropanol with 5-HMF probably competes for Lewis acid sites for the property of accepting electrons and adsorbing oxygenates, making isopropanol decompose in acetone and H_2_. The hydrogen is activated and adsorbed by the metallic sites, generating the hydrogenation of the 5-HMF of the aldehyde group -CHO producing BHMF, emphasizing that this hydrogenation is a slow stage. Subsequently, hydrogenolysis is promoted, which is a rapid reaction for the compounds BHMF and MFA to produce 2,5-DMF [18]. Bui et al. described how those ethers are generated by isopropanol in the -OH of 5-HMF from the superficial Lewis acid sites of the catalyst [87]; furthermore, these Lewis acid sites also generate the reductive etherification of carbonyl compounds [88]. Han et al. mentioned that first, the dehydration (−H_2_O) of isopropanol occurs, and later, the electrons of the hydroxyl group of 5-HMF form ethers with ether compound (1), where the aldehyde group is reduced, producing the ether compound (2) which follows hydrogenolysis of the hydroxyl group forming the ether compound (3) and finally by hydrogenolysis the desired product 2,5-DMF. Another route is the hydrogenation of -CHO from 5-HMF to the BHMF compound, which in contact with isopropanol, generates the ether compound (2) and can undergo another formation of the ether compound (4) which, through slow hydrogenolysis reactions, becomes 2,5-DMF [17]. Using ^1^H NMR for these compounds, esterified by isopropanol, some signals were found.

#### 2.2.3. Elucidation by ^1^H NMR of Products and By-Products of the Conversion of 5-HMF to 2,5-DMF

The crude reaction from each hydrogen donor (formic acid and isopropanol) was analyzed by ^1^H NMR to identify the 2,5-DMF products. Figure 17 show the ^1^H NMR spectrum for the monometallic catalyst with nickel mixtures Ni^0^NiOA/TZ3 in 24 h of reaction from the hydrogen donor formic acid. In this spectrum, the singlet formed by the (CH_3_, δ: 2.35 ppm) CH_3_ and (CH, δ: 5.95 ppm) of 2,5-DMF is observed. These signals coincide with those reported by Hu et al. [89]. Some signals of the formate compounds formed by the esterification of formic acid in the hydroxyl group explained above were found. Likewise, a characteristic signal of the BHMF compound reduced from 5-HMF was located. Neither formic acid nor 5-HMF signals were observed, deducing a 99% 5-HMF conversion, as already demonstrated by chromatography.

Figure 18 display the ^1^H NMR spectrum for the monometallic catalyst with nickel mixtures Ni^0^NiOA/TZ3 in 24 h of reaction from the hydrogen donor isopropanol. In this spectrum, the singlet formed by the (CH_3_, δ = 2.18 ppm) CH_3_ and (CH, δ = 5.95 ppm) of 2,5-DMF is revealed. These signals coincide with those reported by Jae et al. [18]. Some signals of the ether compounds formed by the esterification of isopropanol [(1), (2), (3)] were found, signals 1.19 ppm of the doublet of the methyl protons formed by isopropanol esterified in the hydroxyl group explained above. Weak signals were located for MFA shifts resulting from the hydrogenolysis of BHMF. No signals from isopropanol and 5-HMF were observed, deducing a 99% 5-HMF conversion, as demonstrated by chromatography.

#### 2.2.4. Recycling Tests of the Ni^0^NiO/ATZ3_WI_ Catalyst

The stability of the Ni^0^NiO/ATZ3_WI_ monometallic catalyst through the reaction recycles in the production of 2,5-DMF was evaluated. In Figure 19, this catalyst showed excellent stability and could be recycled five times. The yield of 2,5-DMF decreased from 46% to 45% in the second reaction cycle; however, this catalyst stabilized at a 44% yield in the successive (3 to 5) recycling tests.

This catalyst did not generate leaching because no coloration was observed in the crude reaction by the nickel species, as observed in other works. This affirms the excellent stability of this catalyst since, after several recycles, it showed that the nickel species were strongly anchored to the support [90]. Likewise, surface carbon poisoning was not generated in this catalyst, as reported by Miao et al., who observed that the species formed from NiAl_2_O_4_ nickel aluminates by the impregnation method have a high resistance to the formation of coke on the surface of the catalysts [91].

## 3. Materials and Methods

### 3.1. Reagents

All chemicals in this work were of analytical quality and were used without any purification treatment: Aluminum trisecbutoxide C_12_H_27_AlO_3_ (Sigma-Aldrich, St. Louis, MO, USA, ≥97%), Titanium butoxide (IV) Ti[O(CH_2_)_3_CH_3_]_4_ (Sigma-Aldrich, ≥97%), Zirconium Butoxide Zr[OCH_2_CH_2_CH_2_CH_3_]_4_ (Sigma-Aldrich, ≥80%), n-Butanol (JTBaker, Phillipsburg, NJ, USA, ≥99.9%) and Sec-butanol (Sigma-Aldrich, ≥99.5%), Ni(NO_3_)_2_.6H_2_O (Sigma-Aldrich), 5-Hydroxymethylfurfural (Sigma-Aldrich, ≥99.5% purity), 2,5-Dimethylfuran (Sigma-Aldrich, ≥99%), Tetrahydrofuran (THF-HPLC Meyer, Del. Tlahuac, Cdmx, MX, ≥98.8%), Isopropanol (JTBaker, ≥99.5%), formic acid (Merck, Darmstadt, Germany, ≥98%), Dodecane (Sigma-Aldrich, ≥99%) and ultra-pure water (18.2 MΩ cm^−1^) was obtained from a PureLab Option-Q water purification system for the entire experiment.

### 3.2. Materials Preparation

#### 3.2.1. Al_2_O_3_-TiO_2_-ZrO_2_ (ATZX, X = 1,2,3) Supports Preparation by Sol-Gel

Mixed oxides Al_2_O_3_-TiO_2_-ZrO_2_ were synthesized by the sol-gel method at room temperature, varying their composition by weight 33.33/33.33/33.33 wt% named ATZ1, 40/40/20 wt% named ATZ2, 20/40/40 wt% named ATZ3. The following molar ratios were used in order to obtain porous materials: alkoxide/butanol = 1/8 by volume and alkoxide/water = 1/16 by volume, kept under constant stirring for 24 h to obtain the gel. Subsequently, a gel was obtained that was subjected to a drying process in an oven at 120 °C for 48 h. These materials were subjected to a thermal calcination process at 550 °C for 12 h with a heating ramp of 2 °C/min.

#### 3.2.2. Preparation of the Monometallic Ni/ATZX (X = 1, 2, 3) Catalysts by Wet Impregnation (WI)

The impregnation of Ni in the support was carried out using the wet impregnation method at 15 wt% Ni in 5 g of support, preparing a solution of 100 mL of double distilled water with the precursor salt of Ni(NO_3_)_2_.6H_2_O (Sigma-Aldrich). Subsequently said solution was added in a balloon flask where the 5 g of support were added, it was kept stirring in the rotary evaporator for 4 h. Then, it was dried under vacuum in a rotary evaporator at a temperature of 60 °C. Later, it was left in the oven for 12 h at 120 °C. After it was subjected to a calcination process at 2 °C/min in airflow at 400 °C, reduction in H_2_ flow was carried out at 400 °C for 4 h. Three catalysts were prepared by this method, one catalyst for each support synthesized at different ATZ compositions.

### 3.3. Preparation of the Monometallic Ni/ATZX (X = 1, 2, 3) Catalysts by Suspension Method (SM)

In total, 2 g of support were purged in a quartz reactor with nitrogen for 10 min at room temperature, then reduced with H_2_ flow for 3 h at a temperature of 300 °C. After reduction, the sample was left to cool with hydrogen until reaching room temperature, and the reactor was purged with N_2_ for 30 min. The solution of Ni(NO_3_)_2_.6H_2_O (Sigma-Aldrich) was added to 20 mL of distilled water, adding the necessary amount of the salt to obtain 15 wt% of Ni loading. Subsequently, it was purged with N_2_ for 10 min in the degassing area to eliminate traces of oxygen. To carry out the reaction, the nickel solution was aggregated to the reactor where the ATZX support was located, and N_2_ was bubbled for one hour. The prepared monometallic catalyst was dried with H_2_ at room temperature for 12 h. Finally, it was activated with a flow of H_2_ at a temperature of 400 °C for 4 h using a heating ramp of 2 °C/min.

### 3.4. Materials Characterization

The surface areas of the catalysts were measured by physisorption of N_2_ at −196 °C (Praxair 5.0 U.A.P., Mexico) using Micromeritics TriStar II equipment (Micromeritics Instrument Corporation, Norcross, GA, USA). The measurements of reduction of metallic species, metallic dispersion, and quantification of basic sites in the monometallic catalyst were conducted by chemisorption (H_2_-TPR, H_2_-TPD and CO_2_-TPD) using a Bel Japan equipment model Belcat-B equipped with a thermal conductivity detector (TCD). The X-ray diffraction patterns were obtained using a Bruker AXS model D2 Phaser diffractometer for powders (Borken, North Rhine-Westphalia, Germany); a Cu anode was also used and the radiation corresponded to the CuKα transition with a wavelength of λ = 1.5418 Å, in a range of 2θ from 20° to 80°. The metallic plasmons were observed by UV-Vis spectroscopy with solids diffuse reflectance DRS employing a UV-Vis spectrophotometer model SHIMADZU UV-2600 (Shimadzu Co., Kyoto, Japan), analyzing in the region of 200–800 nm using BaSO_4_ as a blank. The techniques of RAMAN spectroscopy were used for the identification of the vibrations of species in the catalyst in an Xplora plus Raman microscope model HORIBA brand (San Francisco, CA, USA) and infrared spectroscopy (FTIR) using an FT-IR equipment model IR Affinity-1 (Fourier Transform Infrared Spectrophotoner, Shimadzu Co., Kyoto, Japan), 120 V~50/60 Hz 150 VA. The morphology and size of the nanoparticles were observed using scanning electron microscopy (SEM) techniques utilizing a JEOL Scanning Electron Microscope (SEM), model JSM-6010LA, and transmission electron microscopy TEM with JEM-2100 (JEOL, Japan) equipment operating at an acceleration voltage of 200 kV. They were analyzed by FTIR-Pyridine, obtaining infrared spectra with Fourier transform using pyridine as a probe molecule to determine the acid properties, in a NICOLET FT-IR equipment model Magna 560 with a resolution of 4 cm^−1^ and 50 scans, which consists of a DTGS detector. X-ray Photoelectron Spectroscopy (XPS) analysis was performed on a SPECS^®^ X-ray photoelectron spectrometer with a PHOIBOS^®^ 150 WAL hemispherical energy analyzer and angular resolution (<0.5°), equipped with an XR 50 X-Ray Al-ray and μ-FOCUS 500 X-Ray monochromator (Al excitation line). The ^1^H NMR (proton nuclear magnetic resonance) spectra were obtained using a Bruker NMR Spectrometer, mod Advance III 600 MHz, employing deuterated chloroform (CDCl_3_, 25 °C).

### 3.5. Catalytic Activity Tests

#### 3.5.1. Catalytic Conversion of 2,5-DMF from 5-HMF

The production of 2,5-DMF from 5-HMF was carried out in a high pressure and temperature stainless steel autoclave (50 mL) with automated heating and a magnetic stirrer. The reaction was performed using 0.2 g of Ni/ATZX catalyst, 0.252 g 5-HMF in a solution of 50 mL of tetrahydrofuran (THF), 0.86 mL (20 mmol) of formic acid (HCOOH), using 0.245 mL of dodecane as internal standard at inert conditions of 20 bars of Arg and a temperature of 210 °C with a constant stirring of 500 rpm (Figure 4).

#### 3.5.2. Analysis of 2,5-DMF by Gas Chromatography (GC)

The reaction samples were taken at different times up to 24 h of reaction. After the reaction, the autoclave was cooled to room temperature, and the solid catalyst was separated from the reaction mixture by filtration. The samples were filtered with a 0.22 µm Nylon syringe filter before being analyzed by Gas Chromatography (GC) in a Shimadzu chromatograph model GC-2010 Plus equipped with an HP-5 19091J-413 capillary column (30 mm × 0.32 mm × 0.25 μm) and an FID detector. The yield of 2,5-DMF was calculated in this manner:(1)$$2,5\text{-}\mathrm{DMF}\;Yield\;\left(\%\right)=\frac{Moles\;of\;2,5\text{-}\mathrm{DMF}\;in\;products}{initial\;moles\;of\;5\text{-}\mathrm{HMF}}\times 100\;$$

#### 3.5.3. Analysis of 5-HMF Conversion by High-Performance Liquid Chromatography (HPLC)

The 5-HMF conversion was measured in a Shimadzu chromatograph model Prominence HPLC liquid equipped with a Restek column model FORCE C18 made of stainless steel with measurements of 250 mm long, 4.6 mm internal diameter and 5 mm particle size. A UV-Vis detector was used at a wavelength of 284 nm, using methanol at a ratio of methanol/water (80/20) as solvent. The HPLC operating conditions were: oven temperature of 30 °C, flow rate of 1 mL·min^−1^ and an injection volume of 5 µL. The conversion of 5-HMF was calculated as follows:(2)$$5\text{-}\mathrm{HMF}\;Conversion\;\left(\%\right)\;:\;\;\frac{moles\;of\;HMF\;produced\;}{initial\;moles\;of\;HMF}\times 100$$

#### 3.5.4. Analysis of HMF Reaction Products by ^1^H NMR Spectroscopy

Once the reaction was completed, the catalyst was separated, taking advantage of its magnetic properties, decanting it from the liquid sample and allowing the solvent (THF) to evaporate at room temperature. The crude from the final sample was analyzed by ^1^H NMR to identify products such as 2,5-DMF and by-products generated in the evaluation. For this, a Bruker NMR Spectrometer, mod Advance III 600 MHz, was employed; deuterated chloroform (CDCl_3_, 25 °C) was used, and the spectra were processed with the MestReNova program. ^1^H NMR of 2,5-DMF was obtained through the catalytic reaction, applying formic acid (600 MHz, CDCl_3_) CH_3_ (δ: 2.35 ppm) and (CH, δ: 5.95 ppm) as hydrogen donors. Isopropanol was also used as hydrogen donor (600 MHz, CDCl_3_) CH_3_ (δ: 2.18 ppm) and (CH, δ: 5.95 ppm).

## 4. Conclusions

Ni/ATZX monometallic catalysts prepared by Wet impregnation (WI) and Suspension method (SM) with 15 wt% Ni prepared by two different synthesis methods exhibited distinct nickel phases Ni^0^, Ni^0^NiO and NiO. The catalyst with Ni^0^NiO/ATZ3_WI_ phase mixtures was more efficient with significant activity for the hydrogenation of 5-HMF, showing a 46% yield towards 2,5-DMF with formic acid as a hydrogen donor. Due to the synergy that exists between acidic sites (Ni^2+^) and metal sites (Ni^0^) on the catalyst surface, a good selectivity for 2,5-DMF was achieved. Good results were obtained in the catalytic evaluation of this catalyst by the catalytic transfer of hydrogen (CTH), concluding that formic acid is an excellent hydrogen donor compared to isopropanol for the efficient synthesis of 2,5-DMF from 5-HMF. Formic acid is of extreme importance because it can be generated from the rehydration of 5-HMF from lignocellulosic biomass. This work provides a novel strategy to develop low-cost, high-performance catalysts based on non-noble metals for second-generation biofuel processes.

## Data Availability

Not applicable.

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
