# Peer review of "2,5-Dimethylfuran Production by Catalytic Hydrogenation of 5-Hydroxymethylfurfural Using Ni Supported on Al2O3-TiO2-ZrO2 Prepared by Sol-Gel Method: The Effect of Hydrogen Donors"

_molecules, 2022, doi:10.3390/molecules27134187_

Round 1
Reviewer 1 Report
The work reports the preparation of 2,5-dimethylfuran (DMF), a promising liquid fuel and chemical feedstock, starting from cellulose-derived 5-(hydroxymethyl)furfural. First of all, this is a relatively well-documented area of research with hundreds of papers reporting the preparation of DMF from HMF under catalytic hydrogenation condition. The authors have prepeared a novel Ni-based catalyst on mixed metal oxide support, characterized, and hydrogenated purified HMF using hydrogen donors like 2-propanol and formic acid. The manuscript needs some level of English editing since at times the science is not clear due to poor choice of works or sentence construction. For example, "It is reported that 0.245 mL of dodecane has been used for a molecule of 5-HMF to 2,5-DMF". What is meant by 'for a molecule'? This sentence "Therefore, 2.5 DMF is chemically stable and, by being insoluble in water, it prevents the absorption of humidity from 75 the environment and the decrease of its performance in the engine." may be simplified for clearer understanding. This sentence "Additionally, compared to ethanol, it is easier to mix with gasoline and more difficult to dissolve in water." may be rephrased. Scheme 2 says that DMF to 2,5-hexanedione requires hydrogenation, which is not true. Just hydrolysis of DMF forms 2,5-hexanedione. This sentence in abstract "Ni/ATZ catalysts have great properties to generate good catalytic activity obtaining yields of 46% 39 through catalytic hydrogenation from 5-HMF to 2.5-DMF, at a reaction temperature of 210oC and 40 20 bar of argon pressure (PAr) with hydrogen donors (FA, 20 mmol) in Tetrahydrofuran (THF) for 41 24 h of reaction." may be simplified for better understanding. The effect of dodecane as internal standard on the yield of DMF is not clear. This sentence in the conclusion " formic acid as hydrogen donor because it provides 3 hydrogen molecules (3H2)" is confusing. How does a molecule of formic acid produces 3 molecules of H2? The catalyst deactivation mechanisms and recyclability studies should be highlighted.
Author Response
Villahermosa Tab., México; June 15, 2022
Dear Reviewer, 1:
Thank a lot for their recommendations. They were an excellent guideline to improve the manuscript.
Open Review
The work reports the preparation of 2,5-dimethylfuran (DMF), a promising liquid fuel and chemical feedstock, starting from cellulose-derived 5-(hydroxymethyl)furfural. First of all, this is a relatively well-documented area of research with hundreds of papers reporting the preparation of DMF from HMF under catalytic hydrogenation condition. The authors have prepared a novel Ni-based catalyst on mixed metal oxide support, characterized, and hydrogenated purified HMF using hydrogen donors like 2-propanol and formic acid.
Question 1) The manuscript needs some level of English editing since at times the science is not clear due to poor choice of works or sentence construction. For example, "It is reported that 0.245 mL of dodecane has been used for a molecule of 5-HMF to 2,5-DMF". What is meant by 'for a molecule'? This sentence "Therefore, 2.5 DMF is chemically stable and, by being insoluble in water, it prevents the absorption of humidity from the environment and the decrease of its performance in the engine." may be simplified for clearer understanding. This sentence "Additionally, compared to ethanol, it is easier to mix with gasoline and more difficult to dissolve in water." may be rephrased.
Response 1): The sentence was rewrite as follow:
Additional advantage of 2.5 DMF is that it is insoluble in water, this property would prevent contamination due to the absorption of humidity from the environment since this absorption of humidity would generate a decrease in engine performance…………….
2,5-DMF can also be used as an additive to improve the quality and efficiency of gasoline, therefore, a small ratio between the mixture of the additive compared to ethanol could be used to optimize the performance of the gasoline engine and release fewer pollutants.
Question 2) Scheme 2 says that DMF to 2,5-hexanedione requires hydrogenation, which is not true. Just hydrolysis of DMF forms 2,5-hexanedione.
Response 2): In Scheme 2 the H2 was removed from the 2,5-DMF reaction to 2,5-hexanedione.
Question 3) This sentence in abstract "Ni/ATZ catalysts have great properties to generate good catalytic activity obtaining yields of 46% 39 through catalytic hydrogenation from 5-HMF to 2.5-DMF, at a reaction temperature of 210oC and 40 20 bar of argon pressure (PAr) with hydrogen donors (FA, 20 mmol) in Tetrahydrofuran (THF) for 41 24 h of reaction." may be simplified for better understanding. The effect of dodecane as internal standard on the yield of DMF is not clear.
Response 3): The abstract was rewritten to adjust the number of words at 198, as follow:
Catalytic hydrogenation reaction of 5-hydroxymethylfurfural (5-HMF) from lignocellulosic biomass was investigated to obtain versatile liquid biofuels such as 2.5-dimethylfuran (2.5-DMF). Catalytic hydrogenation from 5-hydroxymethylfurfural (5-HMF) to 2.5-dimethylfuran (2.5-DMF) was thoroughly studied on the metal nickel catalysts supported on Al2O3-TiO2-ZrO2 (Ni/ATZ) mixed oxides using isopropanol and formic acid (FA) as hydrogen donors. The structural characteristics of the materials were studied using different physicochemical methods including N2 physisorption, XRD, RAMAN, DRS UV-Vis, FT-IR, SEM, FT-IR Pyad , H2-TPD, CO2-TPD, H2-TPR, TEM and XPS. The influence of metal content (wt%), Ni particle size (nm), Nickel Ni0, Ni0/NiO and NiO species, metal active sites and acid-based sites on the catalyst surface and the effect of the hydrogen donor (isopropanol and formic acid) were systematically studied. It was shown that the Ni0NiO/ATZ3WI catalyst synthesized by the impregnation method (WI) generated a good synergistic effect between the species, facilitating a good catalytic hydrogenation on the conversion of 5-HMF to 2,5-DMF under optimal reaction conditions at a temperature of 210 ° C and 20 bar pressure of Argon (Ar) using formic acid as hydrogen donor for 24 hours of reaction. Second generation 2.5-DMF biofuel and 5-HMF conversion by-products were analyzed and elucidated using 1H NMR.
Question 4) This sentence in the conclusion " formic acid as hydrogen donor because it provides 3 hydrogen molecules (3H2)" is confusing. How does a molecule of formic acid produces 3 molecules of H2?
Response 4): The conclusion was rewritten, as follow:
Ni/ATZX monometallic catalysts prepared by Wet impregnation (WI) and Suspension method (SM) with 15 wt% Ni prepared by two different synthesis methods exhibited distinct nickel phases Ni0, Ni0NiO and NiO. The catalyst with Ni0NiO/ATZ3WI phase mixtures was more efficient with significant activity for the hydrogenation of 5-HMF showing a 46% yield towards 2,5-DMF with formic acid as hydrogen donor. Due to the synergy that exists between acidic sites (Ni2+) and metal sites (Ni0) on the catalyst surface, a good selectivity for 2,5-DMF was achieved. Good results were obtained in the catalytic evaluation of this catalyst by catalytic transfer of hydrogen (CTH), concluding that formic acid is an excellent hydrogen donor compared to isopropanol for an efficient synthesis of 2,5-DMF from 5- HMF. Formic acid is of extreme importance because it can be generated from the rehydration of 5-HMF from lignocellulosic biomass. This work provides a novel strategy to develop low-cost, high-performance catalysts based on non-noble metals for second-generation biofuel processes.
Question 5) The catalyst deactivation mechanisms and recyclability studies should be highlighted.
Response 5): section 2.2.4 was added.
2.2.4. Recycling tests of the Ni0NiO/ATZ3WI catalyst.
NOTE: Additionally, we corrected several typos error and rewritten some paragraphs and sentences. The quality of the figures and the size of letters were improved for a better visualization
We indicated all changes in the manuscript with red letter.
Sincerely Yours,
The authors
Reviewer 2 Report
The study on the synthesis of 2,5-DMF is very interesting for the readers of Molecules, but before the acceptance major revisions are necessary:
1) the abstract is too long, some sentences can be unified and, in my opinion, the results could be avoided.
2) In all the manuscript the references should be appropriately written (as example, line 67 lacks of references as well as lines 94, 97 and so on)
3) line 139 It is not clear if low selectivity of 2,5-DMF occurs with low pressure of H2, please, clearly specify
4) pg 4 the scheme of involved organic reactions is really too complicate and the sentences are too small , please simplify
5)line 159 and the paragraph of the material characterization: It is NOT clear which catalyst is prepared by impregnation or by SM (please, specify SM as line 174)
6)in the introduction avoid to write the conclusions (line 165)
7) Table 1 It is NOT clear which synthetic method is used for each material
8) Several characterizations are described but it is not clear the results if is not clear the synthetic difference between the catalysts
9) in the catalytic tests (line 530) the authors should, first, describe the organic reactions before the comments to the figure.
10)Experimental : please, clearly specify the precursor of oxides
I think that the manuscript should be revisited, in this form it is not clear
Author Response
Villahermosa Tab., México; June 15, 2022
Dear Reviewer, 2:
Thank a lot for their recommendations. They were an excellent guideline to improve the manuscript.
Open Review
The study on the synthesis of 2,5-DMF is very interesting for the readers of Molecules, but before the acceptance major revisions are necessary.
Question 1) the abstract is too long, some sentences can be unified, and, in my opinion, the results could be avoided.
Response 1) The abstract was rewrite. The number of words were adjusted to 198 and the results were mentioned briefly.
Question 2) In all the manuscript the references should be appropriately written (as example, line 67 lacks of references as well as lines 94, 97 and so on)
Response 2) The references were added, and the texts of the above-mentioned lines were reformulated.
Question 3) line 139 It is not clear if low selectivity of 2,5-DMF occurs with low pressure of H2, please, clearly specify
Response 3) The final sentence was rewrite as: It is reported that, in the reactions of 5-HMF with H2 pressure, a lower yield and selectivity for 2,5-DMF is obtained; because it causes the additional reduction of the furan ring producing dimethyltetrahydrofuran (DMTHF), these are the main problems in the selective hydrogenation of 5-HMF to 2,5-DMF
Question 4) pg 4 the scheme of involved organic reactions is really too complicate and the sentences are too small , please simplify
Response 4) Organic molecules in scheme 1 have been edited and the nomenclature of the molecules has been added in the caption of the scheme for better visualization.
Question 5) line 159 and the paragraph of the material characterization: It is NOT clear which catalyst is prepared by impregnation or by SM (please, specify SM as line 174).
Response 5) The sentence of the line 159 was corrected as the line 174.
Line 159: In this work Ni/ATZ catalysts prepared by Wet Impregnation (WI) and Suspension Method (SM) and characterized using several characterization techniques.
Line 174: The values obtained (SBET, PD y VBJH) of the supports and monometallic catalysts synthesized by Wet Impregnation (WI) and Suspension Method (SM) are shown in Table 1,
Question 6) in the introduction avoid to write the conclusions (line 165)
Response 6) This sentence was eliminated of the introduction.
The catalyst with Ni0NiO/ATZ3 phase mixtures was more efficient with significant activity for the hydrogenation of 5-HMF showing a 46% yield towards 2,5-DMF with formic acid as hydrogen donor.
Question 7) Table 1 It is NOT clear which synthetic method is used for each material
Response 7) All images and tables were edited with the synthesis method used in the catalysts by wet Impregnation (WI) and suspension method (SM) to have a better understanding.
Question 8) Several characterizations are described but it is not clear the results if is not clear the synthetic difference between the catalysts
Response 8) All images and tables were edited with the synthesis method used in the catalysts by wet Impregnation (WI) and suspension method (SM) to have a better understanding.
Question 9) in the catalytic tests (line 530) the authors should, first, describe the organic reactions before the comments to the figure.
Response 9) The organic reactions are described in the scheme 1 and 3 of the paper.
Question 10) Experimental : please, clearly specify the precursor of oxides
Response 10) In the section 3.1 the reagents are mentioned and in the section 3.2 sol-gel method of alkoxides and alcohols are described.
- Materials and Methods
3.1. Reagents
All chemicals in this work were of analytical quality and were used without any purification treatment, Aluminum trisecbutoxide C12H27AlO3 (Sigma-Aldrich, ≥97%), Titanium butoxide (IV) Ti[O(CH2)3CH3]4 (Sigma-Aldrich, ≥97%), Zirconium Butoxide Zr[OCH2CH2CH2CH3]4 (Sigma-Aldrich, ≥ 80%), n-Butanol (JTBaker, ≥99.9%) and Sec-butanol (Sigma-Aldrich, ≥99.5%), Ni(NO3)2.6H2O (Sigma-Aldrich), 5-Hydroxymethylfurfural (Sigma-Aldrich, ≥99.5% purity), 2,5- Dimethylfuran (Sigma-Aldrich, ≥99%), Tetrahydrofuran (THF-HPLC Meyer, ≥98.8%), Isopropanol (JTBaker, ≥99.5%), formic acid (Merck, ≥98%), Dodecane (Sigma-Aldrich, ≥99%), ultra-pure water (18.2 MΩ cm−1) obtained from a PureLab Option-Q water purification system for the entire experiment.
3.2. Materials preparation
3.2.1. Al2O3-TiO2-ZrO2 (ATZX, X=1,2,3) supports preparation by Sol-Gel
NOTE: Additionally, we corrected several typos error and rewritten some paragraphs and sentences. The quality of the figures and the size of letters were improved for a better visualization
We indicated all changes in the manuscript with red letter.
Sincerely Yours,
The authors
Round 2
Reviewer 2 Report
In this second review, afrte the suggestions of all the reviewers, the manuscript was improved, so it can be published in this form